# The exon junction complex regulates the splicing of cell polarity gene *dlg1* to control Wingless signaling in development

**Min Liu[1,2,3†], Yajuan Li[1,2†], Aiguo Liu[1,2,3], Ruifeng Li[2,3,4,5], Ying Su[2‡], Juan Du[1,2], Cheng Li[2,3,4,5], Alan Jian Zhu[1,2,3*]**

[1]State Key Laboratory of Membrane Biology and Minstry of Education Key Laboratory of Cell Proliferation and Differentiation, Peking University, Beijing, China; [2]School of Life Sciences, Peking University, Beijing, China; [3]Peking-Tsinghua Center for Life Sciences, Academy for Advanced Interdisplinary Studies, Peking University, Beijing, China; [4]Center for Bioinformatics, Peking University, Beijing, China; [5]Center for Statistical Science, Peking University, Beijing, China

**Abstract** Wingless (Wg)/Wnt signaling is conserved in all metazoan animals and plays critical roles in development. The Wg/Wnt morphogen reception is essential for signal activation, whose activity is mediated through the receptor complex and a scaffold protein Dishevelled (Dsh). We report here that the exon junction complex (EJC) activity is indispensable for Wg signaling by maintaining an appropriate level of Dsh protein for Wg ligand reception in *Drosophila*. Transcriptome analyses in *Drosophila* wing imaginal discs indicate that the EJC controls the splicing of the cell polarity gene *discs large 1 (dlg1)*, whose coding protein directly interacts with Dsh. Genetic and biochemical experiments demonstrate that Dlg1 protein acts independently from its role in cell polarity to protect Dsh protein from lysosomal degradation. More importantly, human orthologous Dlg protein is sufficient to promote Dvl protein stabilization and Wnt signaling activity, thus revealing a conserved regulatory mechanism of Wg/Wnt signaling by Dlg and EJC.

*For correspondence: zhua@pku.edu.cn

†These authors contributed equally to this work

Present address: ‡Institute of Evolution and Marine Biodiversity, Ocean University of China, Qingdao, China

Competing interests: The authors declare that no competing interests exist.

## Introduction

Canonical Wingless (Wg)/Wnt signaling plays an evolutionarily conserved role in dictating cell proliferation, pattern formation, stem cell maintenance and adult tissue homeostasis. Given the importance of Wg/Wnt signaling in many cellular processes, it is not surprising that dysregulation of Wg/Wnt signaling in humans results in developmental defects as well as cancer (*MacDonald et al., 2009*; *Clevers and Nusse, 2012*). In *Drosophila*, Wg ligand binds to the seven-pass transmembrane receptors Frizzled/Frizzled2 (Fz/Fz2) and a co-receptor Arrow (Arr), a homolog of vertebrate LRP5/6. Formation of this trimeric complex activates a scaffold protein Dishevelled (Dsh) on the plasma membrane, leading to disruption of the Axin-mediated degradation complex and hence stabilization of Armadillo (Arm), a homolog of vertebrate β-catenin. Accumulated Arm then translocates to the nucleus to activate target gene transcription (*MacDonald and He, 2012*). Although core components of the Wg/Wnt signaling cascade have been identified, gaps in the understanding of this critical signaling pathway still remain to be filled. To unveil novel regulators of Wg signaling, we conducted a genome-wide RNAi screen in the developing *Drosophila* wing, from which a RNA binding exon junction complex (EJC) emerged as a positive regulator of Wg signaling.

The EJC is known to act in several aspects of posttranscriptional regulation, including mRNA localization, translation and degradation (*Tange et al., 2004*; *Le Hir et al., 2016*). After transcription, the pre-mRNA associated subunit eIF4AIII is loaded to nascent transcripts about 20–24 bases

**eLife digest** Animal development involves different signaling pathways that coordinate complex behaviors of the cells, such as changes in cell number or cell shape. One such pathway involves a protein called Wingless/Wnt, which controls cell fate and growth and is also involved in tumor formation in humans. In recent decades, scientists have made a lot of progress in understanding how this signaling pathway operates. However, it is not well understood how the Wingless/Wnt signaling pathway interacts with other regulatory networks during development.

Now, Liu, Li et al. unveil a new regulatory network that controls the Wingless/Wnt pathway in fruit flies and in mammalian cells grown in the laboratory. The experiments show that an RNA binding protein family named the Exon Junction Complex positively regulates a protein called Dishevelled, which serves as a hub in the Wingless/Wnt pathway. The Exon Junction Complex keeps the amount of Dishevelled protein in check via an interaction with another protein referred to as Discs large. Further experiments indicated that Discs large binds to and protects Dishevelled from being degraded inside the cell.

Liu et al.'s findings highlight a new control mechanism for the Wingless/Wnt signaling pathway. In the future, the findings may also aid the development of new approaches to prevent or treat birth defects and cancer.

upstream of each exon junction, resulting in binding of Mago nashi (Mago)/Magoh and Tsunagi (Tsu)/Y14 proteins to form the pre-EJC core complex. The pre-EJC then recruits other proteins including Barentsz (Btz) to facilitate its diverse function (*Shibuya et al., 2004*). In vertebrates, the EJC is known to ensure translation efficiency (*Nott et al., 2004*) as well as to activate nonsense-mediated mRNA decay (NMD) (*Gatfield et al., 2003*; *Chang et al., 2007*). In *Drosophila*, however, the EJC does not contribute to NMD (*Gatfield et al., 2003*). It is instead required for the *oskar* mRNA localization to the posterior pole of the oocyte (*Newmark and Boswell, 1994*; *Hachet and Ephrussi, 2001*; *Mohr et al., 2001*; *van Eeden et al., 2001*; *Palacios et al., 2004*). Very recently, the pre-EJC has been shown to play an important role in alternative splicing of mRNA in *Drosophila*. Reduced EJC expression results in two forms of aberrant splicing. One is the exon skipping, which occurs in *MAPK* and transcripts that contain long introns or are located at heterochromatin (*Ashton-Beaucage et al., 2010*; *Roignant and Treisman, 2010*). The other is the intron retention on *piwi* transcripts (*Hayashi et al., 2014*; *Malone et al., 2014*). Furthermore, transcriptome analyses in cultured cells indicates the role of EJC in alternative splicing is also conserved in vertebrates (*Wang et al., 2014*).

In this study, we have utilized the developing *Drosophila* wing as an in vivo model system to investigate new mode of regulation of Wg signaling. We find that the pre-EJC positively regulates Wg signaling through its effect on facilitating Wg morphogen reception. Further studies reveal that the basolateral cell polarity gene *discs large 1 (dlg1)* is an in vivo target of the pre-EJC in Wg signaling. We show that Dlg1 acts independently from its role on cell polarity to stabilize Dsh protein, thus allowing Wg protein internalization required for signaling activation. Furthermore, we demonstrate that human Dlg2 exhibits a similar protective role on Dvl proteins to enhance Wnt signaling in cultured human cells. Taken together, our study unveils a conserved regulatory mechanism of the EJC and Dlg in Wg/Wnt signaling.

## Results

### The pre-EJC is required for Wg signaling in the developing *Drosophila* wing

The majority of the Wg/Wnt signaling components have been identified through classical forward genetic screens in *Drosophila* (*Swarup and Verheyen, 2012*; *Jenny and Basler, 2014*). However, these screens failed to uncover a regulatory role of RNA processing in Wg signaling, probably due to the fact that most components of RNA machineries exhibit pleiotropic effects in early development. In an in vivo RNAi screen, we found that knocking down three core components of the pre-

EJC, *tsu*, *mago* and *eIF4AIII*, respectively, resulted in loss of marginal tissues and sensory bristles in adult *Drosophila* wing blade (*Figure 1—figure supplement 1E–G*), which resembles stereotypical phenotypes associated with reduced Wg signaling. Furthermore, loss-of-function *tsu^{Δ18}* or *mago^{93D}* mutants (*Roignant and Treisman, 2010*) displayed similar defects in wing development (*Figure 1A, B*). To confirm that Wg signaling was indeed altered in pre-EJC mutants, we examined in wing imaginal discs the expression of two Wg signaling targets, *senseless* (*sens*) and *Distal-less* (*Dll*) (*Seto and Bellen, 2006*), which respond to graded Wg morphogen. We observed obvious loss of Sens and Dll protein production in *tsu* or *mago* somatic clones (*Figure 1C–F*; *Figure 1—figure supplement 2A–D*). However, Sens expression was not altered in somatic clones of *btz* (*Figure 1—*

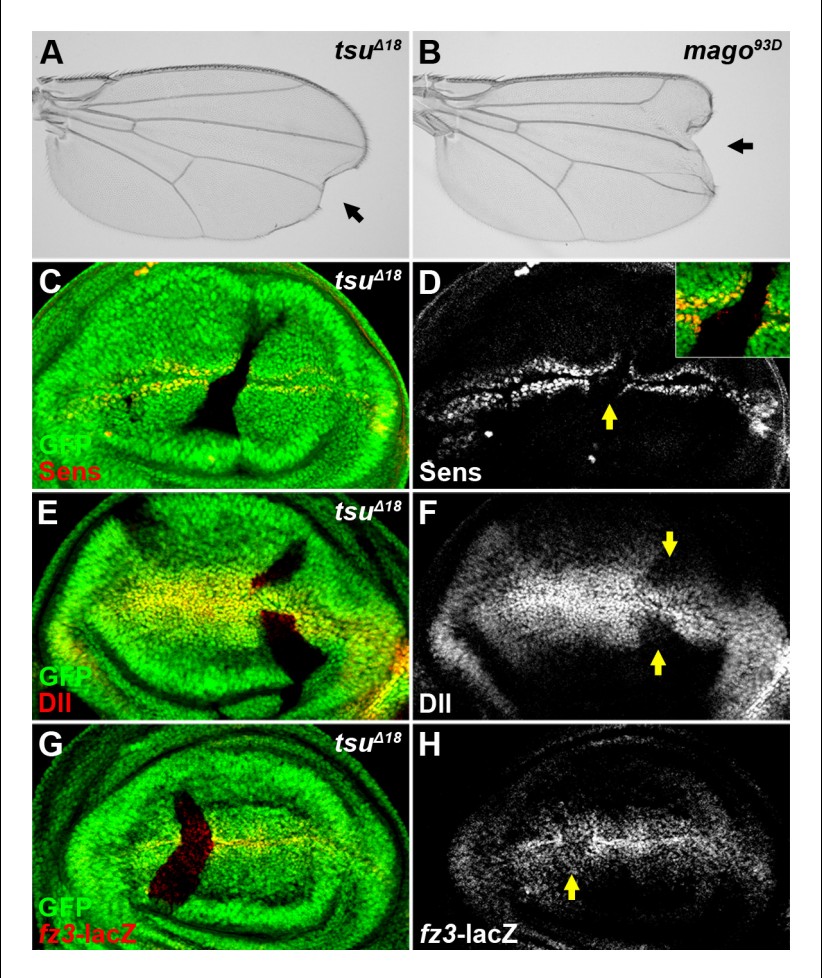

**Figure 1.** The pre-EJC positively regulates Wg signaling. (A,B) A typical loss of Wg signaling wing margin phenotype was observed when *tsu^{Δ18}* or *mago^{93D}* mutant somatic clones were generated in adult wings. Arrows indicate serrated wing margin. (C–H) The production of Wg signaling targets Sens (C,D), Dll (E,F) and *fz3-lacZ* (G, H) was reduced in *tsu^{Δ18}* clones (marked by the absence of GFP and hereafter in subsequent figures). The positions of clones are indicated by arrows.

The following figure supplements are available for figure 1:

**Figure supplement 1.** Knocking down individual components of the pre-EJC reduces Wg signaling in the developing wing.

**Figure supplement 2.** The pre-EJC component Mago positively regulates Wg signaling.

**Figure supplement 3.** The EJC cytoplasmic component Btz does not regulate Wg signaling.

*figure supplement 3A,B*), which is a cytoplasmic component of the EJC (*Palacios et al., 2004*), suggesting that the role of EJC in Wg signaling is independent of its cytoplasmic function. To directly monitor transcriptional activity of Wg signaling in pre-EJC defective wing discs, two *lacZ* enhancer traps inserted in the genomic loci of Wg targets, *frizzled3 (fz3)* (*Sivasankaran et al., 2000*) and *Dll*, were used. As expected, the expression of *fz3-lacZ* and *Dll-lacZ* was decreased when *tsu* activity was reduced (*Figure 1G,H*, *Figure 1—figure supplement 1H–J*). Taken together, the above data indicate that the pre-EJC activity is required for Wg signaling activation in the developing fly wing.

## The pre-EJC regulates Wg signaling at the level of Wg protein reception

To understand how the pre-EJC regulates Wg signaling, we examined the expression of Wg signaling components in wing discs with altered EJC activity. We found that the amount of Wg protein present in *tsu* or *mago*, but not *btz*, mutant clones was significantly increased (*Figure 2A–C*; *Figure 1—figure supplement 2E,F*; *Figure 1—figure supplement 3C,D*). This finding was surprising given the observation that Wg signaling was downregulated in mutant cells. Clue to the understanding of this apparent contradiction came from the observation that Wg signaling was ectopically activated in wildtype cells immediately next to mutant clones (insets of *Figures 1D* and *2C*). This result suggests that increased Wg protein present in mutant clones is sufficient to activate

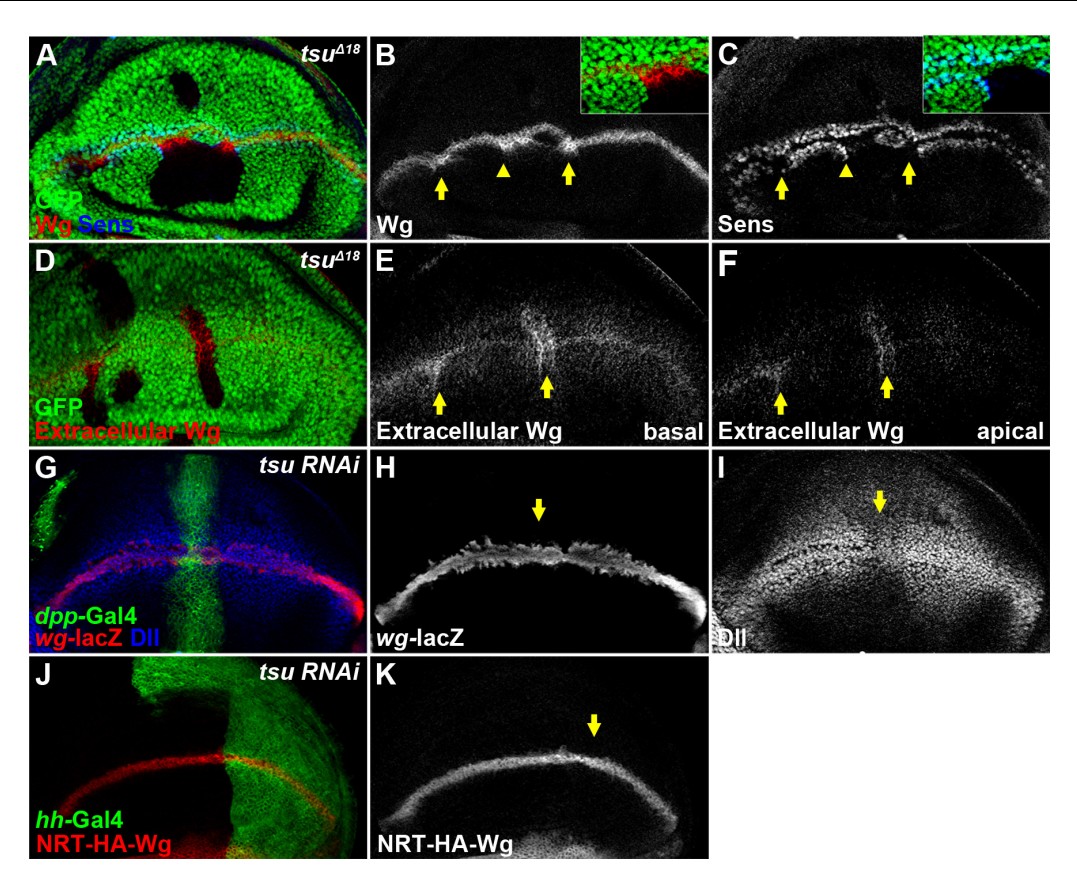

**Figure 2.** The pre-EJC is required for Wg morphogen reception. (**A–C**) Wg protein stained with the conventional method (**B**) was accumulated in *tsu^{Δ18}* clones where the expression of Sens was reduced (arrows). The regions marked by arrowheads are shown in insets. Note that Sens was activated in cells outside the mutant clone (**C**). (**D–F**) Extracellular Wg was accumulated at the basal (**E**) and apical extracellular spaces of the *tsu^{Δ18}* clones (**F**). (**G–I**) The activity of *wg-lacZ* (**H**) did not change when *tsu RNAi* was expressed by *ptc*-Gal4 (marked by GFP and arrows). Note that Dll expression was reduced in *tsu RNAi*-expressing cells (**I**). (**J,K**) The expression of plasma membrane bound NRT-HA-Wg did not change when *tsu RNAi* was expressed by *hh*-Gal4.

signaling in neighboring wildtype cells whilst mutant cells are incapable of receiving Wg input. Therefore, the Wg signaling defects observed in mutant cells could be caused by blockage of Wg reception at the plasma membrane. Consistent with this hypothesis, extracellular Wg protein (*Strigini and Cohen, 2000*) accumulated significantly in both apical and basolateral extracellular spaces (*Figure 2D–F*). Furthermore, we showed that the increased extracellular Wg was not due to heightened *wg* gene transcription nor Wg protein secretion in producing cells because neither the expression of *wg-lacZ* (a *wg* transcription reporter) nor the amount of NRT-Wg (a membrane-tethered form of Wg) was altered (*Figure 2G–K*; also see *Figure 3—figure supplement 3B* for *wg* transcription analyses).

Previous studies have shown that inhibiting Wg protein endocytosis by reducing either the activity of Wg receptor complex Fz2/Arr or the scaffold protein Dsh results in accumulation of Wg protein on the plasma membrane (*Han et al., 2005*). In *tsu* and *mago* mutant clones, we found that the expression of Dsh (*Figure 3A–D*), but not Fz2 or Arr (*Figure 3—figure supplement 1*), was obviously reduced. This result was further verified in cultured *Drosophila* Schneider 2 (S2) cells in which *tsu* or *mago* was knocked down by RNAi (*Figure 3I*). To unveil the functional importance of the EJC regulation on Dsh, overexpressed *dsh* was able to rescue the wing margin defects caused by *tsu* knockdown (*Figure 3H*; cf. *Figure 3E*), whereas *fz2* or *arr* had little effect (*Figure 3F,G*). The above experiments suggest that the pre-EJC acts primarily through Dsh in wing discs to regulate Wg signaling reception.

## The pre-EJC regulates the splicing of cell polarity gene *dlg1*

The pre-EJC is known to function at the level of target mRNA splicing. However, we were not able to correlate reduced Dsh protein production observed in pre-EJC defective cells with decreased *dsh* mRNA expression (*Figure 3J*; *Figure 3—figure supplement 2*; *Figure 3—figure supplement 3A*). This inconsistency could be due to the fact that the *dsh* locus does not contain any intron, and in principle, may not be subjected to the EJC regulation. Thus, we reasoned that the pre-EJC must regulate Wg signaling through a Dsh-interacting protein whose mRNA expression may be controlled by the pre-EJC.

To uncover *bona fide* EJC targets whose encoded protein products interact with Dsh to control Wg signaling reception, we utilized whole transcriptome RNA-seq to compare mRNAs extracted from wildtype (i.e. overexpressing *lacZ*) and pre-EJC-defective (i.e. overexpressing *tsu RNAi*) wing discs, respectively. We found that the expression of 1447 mRNAs was altered by more than 25% when the pre-EJC activity was downregulated (*Supplementary file 2*). This list likely includes both direct as well as indirect pre-EJC targets as 629 genes were identified whose expression was increased. Among those genes whose mRNA levels were reduced we found that the expression of *sens* and *Dll* mRNA, two Wg signaling transcription targets, was decreased by more than 50% in wing discs expressing *tsu RNAi*. Consistent with the RT-PCR result (*Figure 3—figure supplement 3*), *wg* and *dsh* mRNA did not show an obvious change.

To effectively narrow down the pre-EJC targets acting in Wg signaling, we compared our candidate list with annotated information extracted from the FlyBase describing validated protein interactions with Dsh (*Carmena et al., 2006*; *Chung et al., 2009*; *Kaplan and Tolwinski, 2010*; *Varelas et al., 2010*; *Johnston et al., 2013*; *Schertel et al., 2013*; *Strutt et al., 2013*; *Warrington et al., 2013*; *Garcia et al., 2014*). Among 818 mRNAs whose expression was downregulated, we identified three genes, *dlg1*, *lethal (2) giant larvae [l(2)gl]* and *diablo (dbo)*, whose protein products are known to directly interact with Dsh. qPCR and RT-PCR analyses were then performed to confirm *bona fide* Dsh interactors that potentially mediate the pre-EJC activity on Dsh (*Figure 3J*; *Figure 3—figure supplement 3B*). In both cases, *dlg1* was the candidate consistently exhibiting reduced expression in the pre-EJC-defective wing disc cells when compared with that in wildtype cells. These results fit well with RNA-seq analyses that the amount of *dlg1* mRNA was reduced by about 30% when the pre-EJC was dysfunctional. We suspected that reduced *dlg1* expression may be a consequence of altered RNA splicing. Apart from previously reported exon skipping caused by dysfunctional EJC (*Ashton-Beaucage et al., 2010*; *Roignant and Treisman, 2010*), we uncovered two additional aberrant splicing events (*Figure 4—figure supplement 1*). The first event was exon inclusion, which retained exons that were normally efficiently spliced in specific isoforms (*Figure 4—figure supplement 1E*). This event likely increases the usage of splice sites. Indeed the RNA-seq analyses revealed a slightly higher usage of annotated splice sites (i.e. 1.4% higher usage for 5′

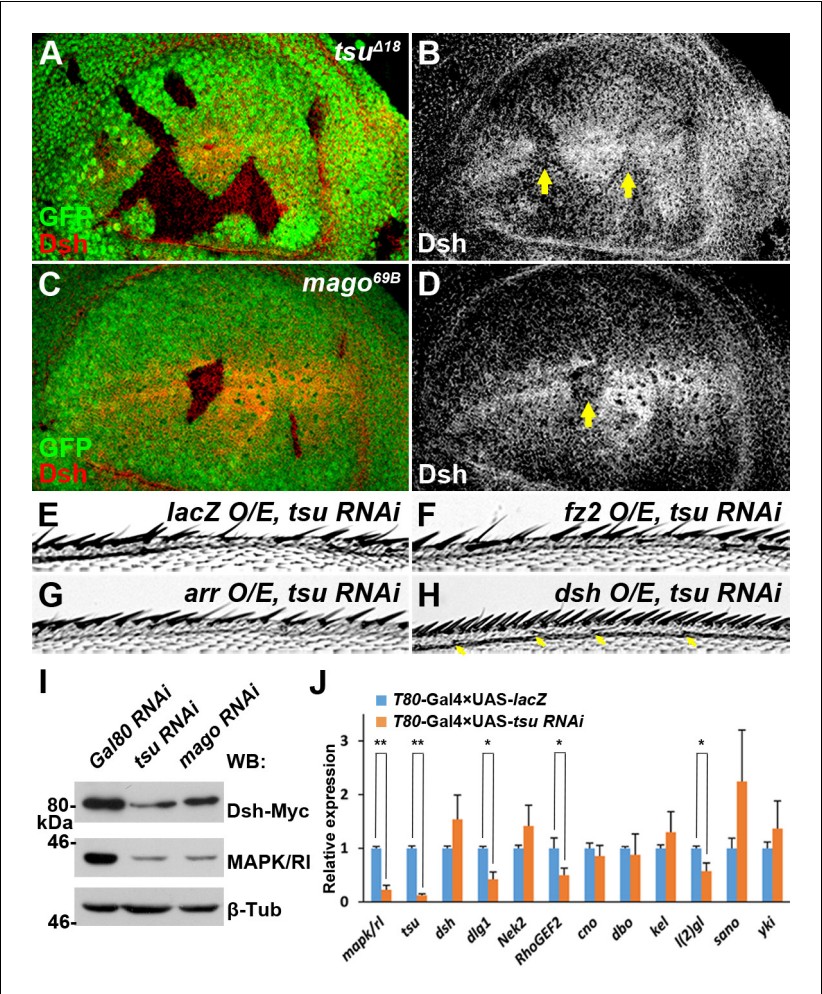

**Figure 3.** The pre-EJC regulates Wg signaling through Dsh. (A–D) The amount of Dsh protein was reduced in $tsu^{\Delta 18}$ and $mago^{69B}$ clones in the wing disc. (E–H) Overexpressing $dsh$ (H), but not $fz2$ (F) or $arr$ (G), rescued the loss of Wg signaling wing margin phenotype caused by $tsu$ knockdown (E). Arrows indicate sensory bristles along the wing margin. (I) The production of Myc-tagged Dsh was reduced when $tsu$ or $mago$ dsRNA was expressed in S2 cells. Yeast $Gal80$ dsRNA served as a negative control for RNAi treatment. MAPK/Rl, a known pre-EJC target served as a positive control for defective pre-EJC. β-Tubulin was used as a loading control for all experiments. (J) Real time RT-PCR revealed that the abundance of $dlg1$, $RhoGEF2$ and $l(2)gl$, but not $dsh$ mRNA, was reduced when $tsu$ dsRNA was expressed in wing discs. $mapk/rl$ served as a positive control. $\alpha$-Tubulin 84B was used to normalize the amount of cDNA template. The experiments were performed in triplicates, and data were represented as the mean+S.D. (*p<0.05; **p<0.01; Student's t-test).

The following figure supplements are available for figure 3:

**Figure supplement 1.** The pre-EJC does not obviously regulate the expression of Fz2 and Arr.

**Figure supplement 2.** The mRNA abundance of cDNA-derived $dsh$ is not altered when the pre-EJC activity is knocked down in S2 cells.

**Figure supplement 3.** The mRNA abundance of $dlg1$ but not those encoding other Dsh-interacting proteins is reduced when the pre-EJC is knocked down in wing discs.

splice sites and 1.5% for 3' splice sites) in pre-EJC depleted wing disc cells compared with those in wildtype cells (*Figure 4—figure supplement 2A*). The second event utilized previously unidentified splice sites to generate new introns and exons (*Figure 4—figure supplement 1F–H*, *2B, C*; *Supplementary file 3*). Together, our bioinformatic analyses suggest that the pre-EJC plays a critical role in alternative splicing by preserving correct usage of splice sites to generate functional mRNA products.

The impact of the pre-EJC regulation on *dlg1* splicing was further verified at the protein level as the amount of Dlg1 protein was reduced in *tsu* mutant wing disc clones (*Figure 4A,B*) as well as in S2 cells treated with dsRNA against individual components of the pre-EJC (*Figure 4E*). In contrast, the level of Dlg1 protein derived from a cDNA expression construct did not change (*Figure 4F*). Furthermore, we demonstrated that reduced *dlg1* expression led to accumulation of extracellular Wg (*Figure 4C,D*), attenuation of Wg signaling in wing discs (*Figure 4—figure supplement 3A–D*), and consequently loss of wing margin and sensory bristles in the adult wing blade (*Figure 4—figure supplement 3E*). Significantly, overexpressed *dlg1* was sufficient to rescue wing margin defects caused by dysfunctional pre-EJC (*Figure 4—figure supplement 3H*). It is known that additional splicing factors are required for the pre-EJC activity (*Reichert et al., 2002*). Inhibiting one of such factors, Rnps1, in the wing disc resulted in the same defect in Dlg1 and Dsh protein production as well as Wg signaling activation (*Figure 4—figure supplement 4*). Together, these experiments indicate that the pre-EJC mediated splicing activity positively regulates *dlg1* to control Wg signaling in *Drosophila*.

## Dlg1 protein stabilizes Dsh in the Wg signal transduction

Dlg1 is important for the maintenance of cell polarity, acting together with L(2)gl to form a basolateral complex to counteract with the apical complex in epithelium (*St Johnston and Ahringer, 2010*). Therefore, it is not surprising that reduced pre-EJC activity led to altered cell polarity in wing disc epithelium (*Figure 4—figure supplement 6*). To investigate if the maintenance of cell polarity is required for Wg signaling, we examined the effects of L(2)gl as well as Bazooka (Baz) and Cdc42 (*Figure 5—figure supplement 1*), two apical complex members (*Harris and Peifer, 2004*; *Warner and Longmore, 2009*), in the Wg signal transduction. Surprisingly, loss of these polarity regulators in wing disc clones did not obviously affect Wg signaling (*Figure 5—figure supplement 2*), implying that the regulation of Wg signaling by Dlg1 may be independent of its function on cell polarity.

Since cell polarity does not contribute significantly to Wg signaling, we reasoned that Dlg1 may directly interact with Dsh to regulate its activity. It has been reported that *Drosophila* Dsh binds in vitro to the K-K-x-x-x-Ψ motif within the I3-insert of the Hook domain in Dlg1 (*Garcia et al., 2014*). We confirmed this interaction in S2 cells as well as in wing discs by immunoprecipitation (*Figure 4G, H*; *Figure 4—figure supplement 6A*). The relevance of such interaction was further illustrated in rescue experiments in which overexpressed *dsh* was able to largely rescue the wing margin defects associated with reduced *dlg1* activity (*Figure 4—figure supplement 3G*).

Next, we investigated how Dlg1 modulates Dsh to facilitate Wg signaling. Knocking down *dlg1* by RNAi in S2 cells led to reduced production of Dsh protein, whilst overexpressing *dlg1* had an opposite effect (*Figure 5A,B*; *Figure 4—figure supplement 5C*). A similar result was observed in vivo when *dlg1* expression was manipulated in wing discs (*Figure 5C–F*). The effect of Dlg1 on Dsh production may be a direct consequence of altered protein stability. Dsh protein ectopically produced in S2 cells exhibited a short half-life of around two hours (*Figure 5—figure supplement 3*). The turnover of Dsh protein could be controlled through the ubiquitin-proteasome or lysosome mediated degradation. We found that the degradation of Dsh protein occurred mainly in lysosome when nascent protein synthesis was blocked by cycloheximide (CHX) (*Figure 5G*). This is consistent with the observation that Dsh colocalized with early endosome and late endosome/lysosome markers in wing discs as well as in S2 cells (*Figure 5H–M*; *Figure 5—figure supplement 4*). Furthermore, heightened *dlg1* expression counteracted with the CHX effect on Dsh degradation (*Figure 5N*). In contrast, Dsh degradation potentiated by *tsu* or *dlg1 RNAi* could be prevented when S2 cells were treated with a lysosomal inhibitor chloroquine (CQ) (*Figure 5O,P*). The above data indicate that the interaction between Dlg1 and Dsh protects Dsh protein from degradation in the lysosome. This conclusion was supported further by the observation that increased *dlg1* expression

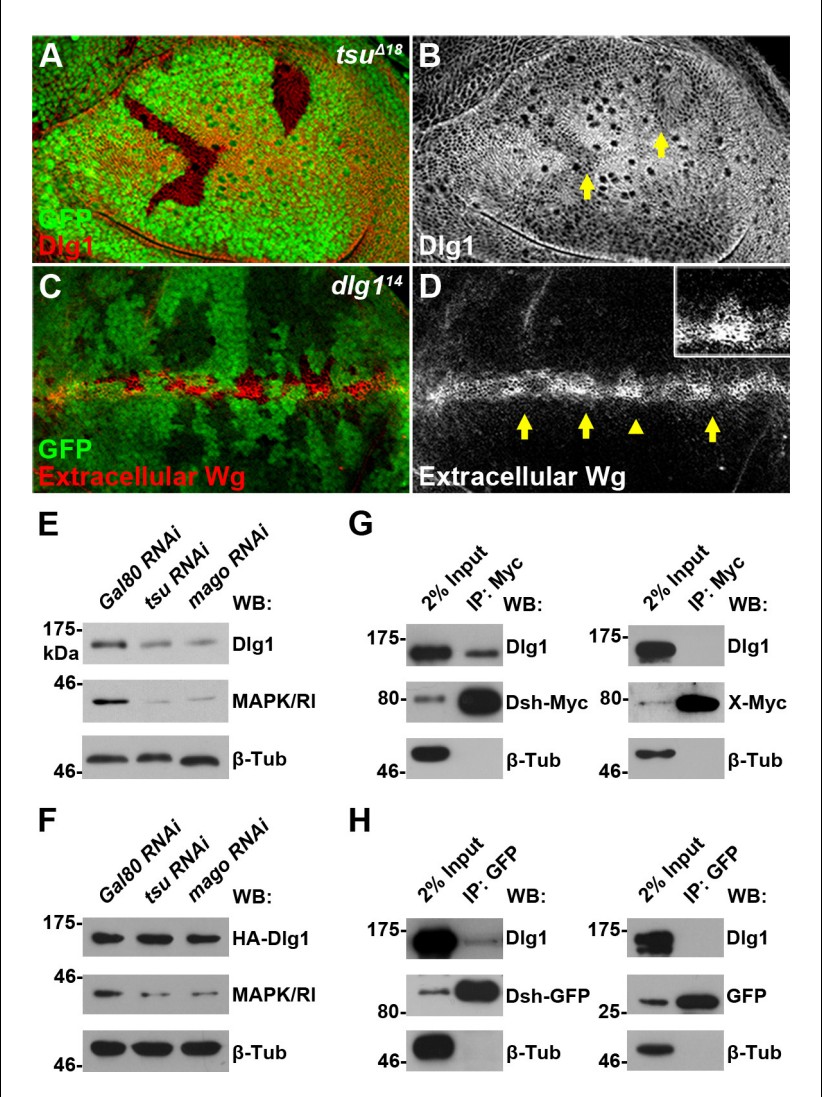

**Figure 4.** The pre-EJC regulates Dsh-interacting protein Dlg1. (A,B) The expression of endogenous Dlg1 protein was reduced in $tsu^{\Delta18}$ clones (indicated by an arrow). (C,D) Extracellular Wg was accumulated in $dlg1^{14}$ clones. Shown in the inset is an enlarged clone with accumulated extracellular Wg. (E,F) The amount of endogenous (E) but not a cDNA-derived Dlg1 protein (F) was reduced when *tsu* or *mago* dsRNA was expressed in S2 cells. MAPK/Rl served as a positive control for dysfunctional pre-EJC activity. (G,H) Endogenous Dlg1 interacted with Myc-tagged Dsh in S2 cells (G) and GFP-tagged Dsh in wing discs expressing the *dsh-gfp* under the control of the *dsh* promoter (H). A Myc-tagged irrelevant protein X or GFP alone served as negative controls, respectively.

The following figure supplements are available for figure 4:

**Figure supplement 1.** The pre-EJC regulates the precise splicing of *dlg1*.

**Figure supplement 2.** The pre-EJC regulates the alternative splicing.

**Figure supplement 3.** Dlg1 is a positive regulator of Wg signaling.

**Figure supplement 4.** Splicing factor Rnps1 regulates Wg signaling.

**Figure supplement 5.** The interaction between Dlg1 and Dsh in S2 cells.

**Figure supplement 6.** The pre-EJC components Mago and Tsu regulate cell polarity in the wing disc.

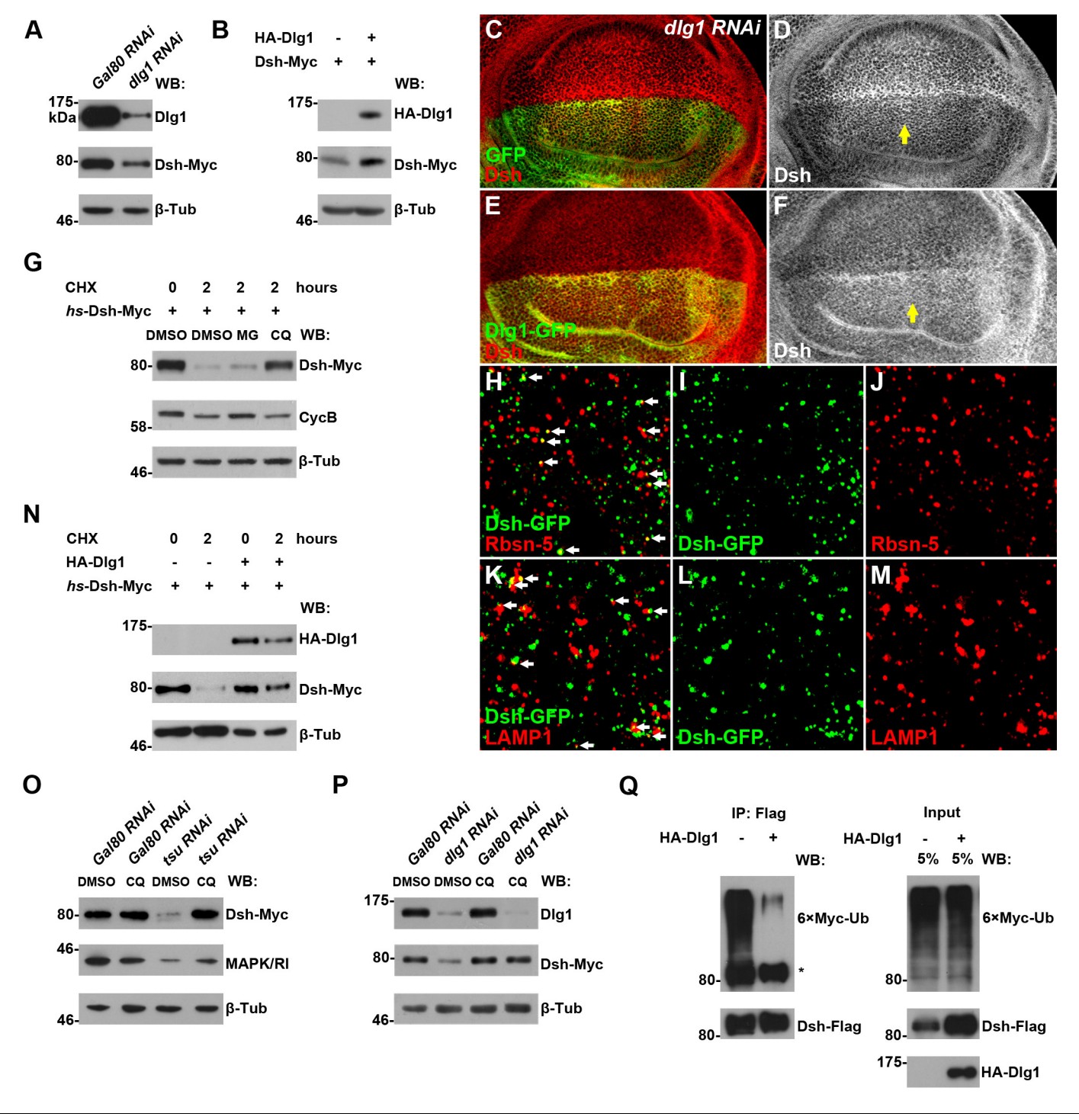

**Figure 5.** Dlg1 regulates Dsh protein turnover. (A) Knocking down *dlg1* by RNAi in S2 cells reduced the amount of Dsh protein. (B) Overexpressing *dlg1* increased the abundance of Dsh protein in S2 cells. Note that Dlg1-PD form was used in all experiments unless mentioned otherwise. (C–F) The amount of Dsh was altered respectively when *dlg1 RNAi* (D) or *dlg1-gfp* (F) was expressed in wing discs by *ap*-Gal4. (G) CHX treatment-induced Dsh protein degradation was inhibited by lysosome inhibitor chloroquine (CQ) but not by proteasome inhibitor MG132 (MG). Cyclin B is known to be degraded in the proteasome, which served as a positive control for MG treatment (*Zhang et al., 2014*). DMSO served as a negative control. (H–M) Dsh protein was detected in endocytic compartments in wing discs expressing *Ubpy* RNAi to prevent lysosome function. Over 10% of GFP-tagged Dsh (*dsh-gfp* under the control of the *dsh* promoter) colocalized with early endosome protein Rbsn-5 [H–J; n (field of view) = 8] and late endosome/lysosome protein marker LAMP1 in wing discs [K–M; n (field of view) = 10]. (N) CHX treatment-induced Dsh degradation was inhibited when *dlg1* was

*Figure 5 continued on next page*

*Figure 5 continued*

overexpressed in S2 cells. (**O,P**) Dsh degradation resulted from *tsu RNAi* (**O**) or *dlg1 RNAi* (**P**) treatment was inhibited by lysosome inhibitor CQ. (**Q**) Overproduced Dlg1 reduced the level of ubiquitination of Dsh. Asterisk indicated non-specific signal of Myc antibody reactivity.

The following figure supplements are available for figure 5:

**Figure supplement 1.** Reduced activity of cell polarity determinants *l(2)gl, baz* or *cdc42* result in polarity defects in wing disc cells.

**Figure supplement 2.** Reduced activity of cell polarity determinants *l(2)gl, baz* or *cdc42* does not result in an obvious Wg signaling defect.

**Figure supplement 3.** The turnover of fly Dsh protein was measured in S2 cells treated with CHX followed by a four-hour chase.

**Figure supplement 4.** Dsh colocalizes with endosome and lysosome markers in S2 cells.

correlated with reduced degree of ubiquitination modification on Dsh, a form of posttranslational modifications required for protein degradation (*Figure 5Q*).

## The regulation of Dvl proteins by Dlg is conserved in vertebrate cells

Consistent with a role of Dlg1 on Dsh stability in *Drosophila* wing development, regulated protein degradation serves as a control mechanism on Dvl protein homeostasis in cultured vertebrate cells (*Gao and Chen, 2010*). It is thus likely that Dlg1 orthologs could act in a similar manner to control Dvl protein stability in vertebrates. Human cells encode five orthologous Dlg proteins, Dlg1-5. However, only Dlg2 contains an intact K-K-x-x-x-Ψ motif required for Dsh interaction as elucidated in *Drosophila* (*Figure 6A*). Indeed, overproduced Dlg2, but not Dlg3, was sufficient to stabilize Dvl proteins in HEK293T cells (*Figure 6B,E*, *Figure 6—figure supplement 1A,B*, *Figure 6—figure supplement 2*). To further illustrate the importance of the K-K-x-x-x-Ψ motif of Dlg2 in stabilizing Dvl proteins, we generated a mutant form of Dlg2 (hDlg2KKAA) where two key lysine residues of the Hook domain were mutated to alanines. As predicted, this Dlg2 mutant failed to stabilize Dvl due to its inability to interact with Dvl (*Figure 6C,D*). Dlg1 and Dlg2 share extensive homology within the Hook domain except that an amino acid is missing in Dlg1's K-K-x-x-x-Ψ motif (*Figure 6A*). We replaced this motif in Dlg2 with amino acids found in Dlg1 (*Figure 6—figure supplement 1C*). The resulting Dlg2SFI-NL mutant showed reduced interaction with Dvl and no effect on Dvl protein stabilization (*Figure 6—figure supplement 1D,E*). To further demonstrate functional consequence of Dlg2 regulation on Dvl protein stability in Wnt signaling, we performed TOPFlash Luciferase Wnt signaling reporter assay in HEK293T cells. Overexpressed wildtype but not the KKAA mutant form of *dlg2* enhanced further the Dvl-induced TOPFlash reporter activity (*Figure 6F*). The above experiments together indicate that the Dlg activity on Dvl protein homeostasis is conserved from flies to vertebrates.

## Discussion

In summary, our study uncovers a specific role of the RNA binding protein complex EJC in the *Drosophila* wing morphogenesis. Our genetic and biochemical analyses demonstrate that the pre-EJC is necessary for Wg morphogen reception to activate the signal transduction. The identification of the cell polarity determinant *dlg1* as one of the pre-EJC targets provides mechanistic basis for the pre-EJC regulation of the Wg signaling. Dlg1 controls the stability of the scaffold protein Dsh, which is the hub of the Wg signaling cascade (*Figure 7*). Importantly, this mode of regulation of Dvl by Dlg is conserved from flies to vertebrates.

### The EJC specifically regulates development

The EJC as well as other RNA binding protein complexes are thought to function in a pleiotropic manner. However, our data presented here together with several recent studies argue that RNA regulatory machineries can act specifically on developmental signaling for pattern formation and organogenesis. It has been increasingly recognized that the production, transport or the location of mRNA are subject to precise regulation in Wg/Wnt signaling. For example, apical localization of *wg*

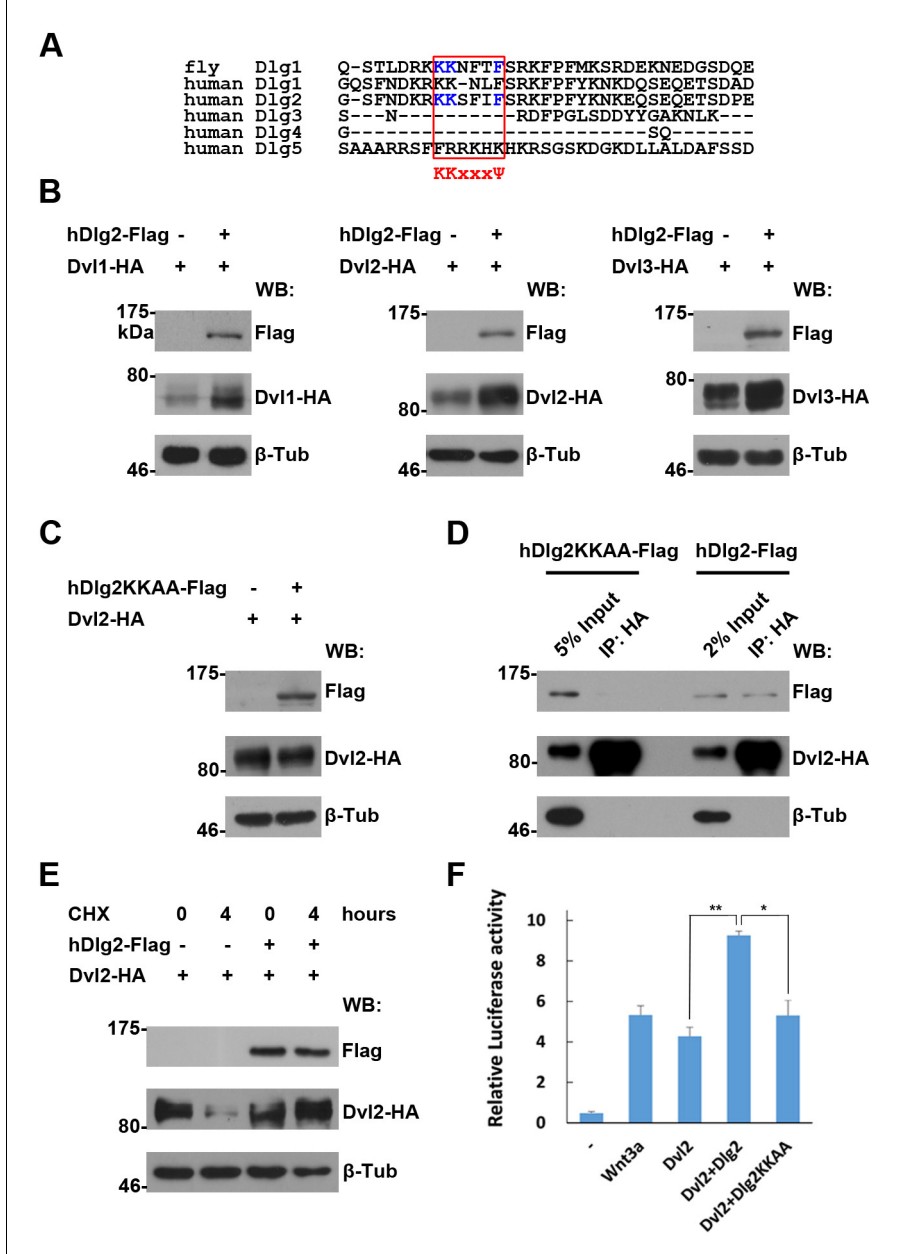

**Figure 6.** The regulation of Dvl proteins by Dlg is conserved in vertebrates. (**A**) Sequence alignment of the I3-insert of Hook domains presented in Dlg proteins reveals sequence conservation between fly Dlg1 and human Dlg orthologs. The K-K-x-x-x-Ψ motif is highlighted in a red box. Amino acids essential for Dsh interaction are shown in blue. (**B,C**) Overproduced wildtype (**B**), but not the KKAA mutant form of hDlg2 (**C**), increased the amount of Dvl proteins in HEK293T cells. (**D**) Dvl protein interacted with wildtype, but not the KKAA mutant form of hDlg2. (**E**) CHX treatment-induced Dvl protein degradation was prevented by overproduced hDlg2. (**F**) Overexpressing wildtype, but not the KKAA mutant form of hDlg2, further increased Dvl2-induced TOPFlash Wnt signaling reporter activity. Ectopic Wnt3a served as a positive control. Experiments were repeated thrice, and data were represented as the mean+S.D. after normalized to *Renilla* activity (**p<0.01; *p<0.05; Student's t-test).

The following figure supplements are available for figure 6:

**Figure supplement 1.** An intact K-K-x-x-x-Ψ motif is required for human Dlg proteins to stabilize Dvl2 in HEK293T cells.

*Figure 6 continued on next page*

*Figure 6 continued*

**Figure supplement 2.** Protein turnover of human Dvl2 protein was measured in HEK293T cells treated with CHX followed by a four-hour chase.

RNA is essential for signal activation in epithelial cells (*Simmonds et al., 2001*; *Wilkie and Davis, 2001*), whilst the RNA binding protein RBM47 regulates Wnt signaling in zebrafish head development (*Guan et al., 2013*) as well as in cancer (*Vanharanta et al., 2014*; *Venugopal et al., 2015*). The specific role of RNA machineries on cell signaling is not limited to Wg/Wnt signaling. It has been reported that RNA-binding protein Quaking specifically binds to the 3'UTR of transcription factor *gli2a* mRNA to modulate Hedgehog signaling in zebrafish muscle development (*Lobbardi et al., 2011*). RNA binding protein RBM5/6 and 10 could differentially control alternative splicing of a negative Notch regulator gene *NUMB*, thus antagonistically regulating the Notch signaling activity for cancer cell proliferation (*Bechara et al., 2013*). Therefore, generally believed pleotropic RNA regulatory machineries emerge as important regulatory means to specifically control cell signaling and related developmental processes.

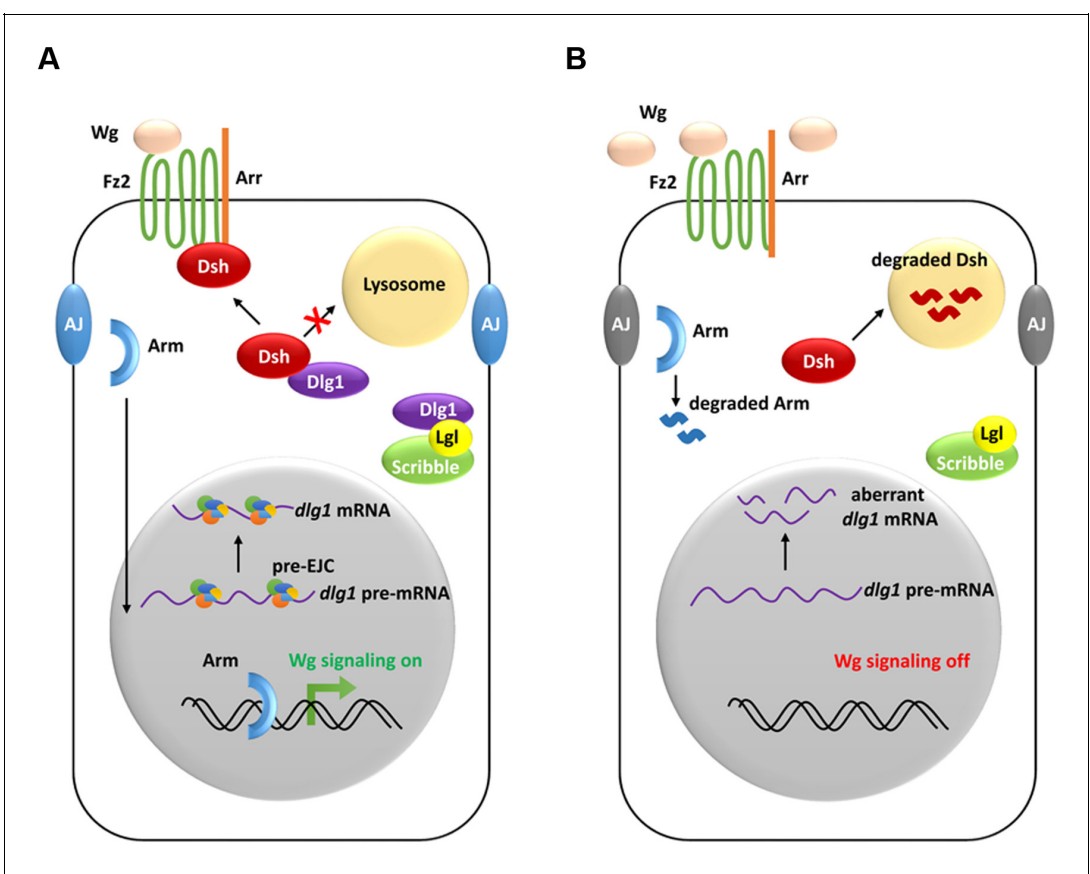

**Figure 7.** A model showing how the pre-EJC functions in the Wg signaling. (**A**) In wildtype wing disc epithelial cells, the pre-EJC activity is required for precise splicing of target transcripts, including *dlg1*. Dlg1 protein functions independently from its role in cell polarity determination to prevent Dsh protein from degradation in the lysosome. Stabilized Dsh works together with Fz2 and Arr, two Wg co-receptors, to facilitate Wg signal reception. Consequently, through a series of cytoplasmic events, Arm protein becomes stabilized, which translocates to the nucleus to activate Wg target gene transcription. (**B**) When the pre-EJC activity was defective in wing disc epithelium, precise splicing of *dlg1* gene is not maintained. Decreased Dlg1 production is not sufficient to stabilize Dsh, resulting in reduced Wg morphogen reception. As a result, Wg is accumulated in the extracellular space of the epithelium, which ultimately fails to protect Arm from degradation, hence reduced Wg signaling.

The most studied function of the EJC in development is to localize *oskar* mRNA to the posterior pole of the oocyte for oocyte polarity establishment and germ cell formation in *Drosophila* (*Newmark and Boswell, 1994*; *Hachet and Ephrussi, 2001*; *Mohr et al., 2001*; *van Eeden et al., 2001*; *Palacios et al., 2004*). Further study suggests that the proper *oskar* RNA localization relies on its mRNA splicing (*Hachet and Ephrussi, 2004*). In light of our study of the EJC activity on *dlg1* mRNA as well as the roles of EJC on *mapk* and *piwi* splicing (*Ashton-Beaucage et al., 2010*; *Roignant and Treisman, 2010*; *Hayashi et al., 2014*; *Malone et al., 2014*), we suspect that EJC might regulate *oskar* mRNA splicing to mediate its mRNA localization. Our RNA-seq analyses identified several hundreds of candidate mRNAs whose expression may be directly or indirectly subjected to EJC regulation. Apart from defects in Wg and MAPK signaling, however, we did not observe altered wing patterning associated with other developmental signaling systems in EJC defective flies, arguing that EJC may primarily regulate Wg and MAPK signaling in patterning the developing wing.

## Regulated Dishevelled activity is required for Wg/Wnt signaling

Wg/Wnt signaling plays a fundamental role in development and tissue homeostasis in both flies and vertebrates. Its activation and maintenance rely on appropriate activity of the ternary receptor complex including Fz family proteins. In *Drosophila*, polarized localization of Fz and Fz2 proteins is essential for activation of non-canonical and canonical Wg signaling, respectively (*Boutros et al., 2000*; *Wu et al., 2004*). Dsh, which acts as a hub mediating both canonical and non-canonical Wg signaling, however, is found at both the apical cell boundary and in the basal side of the cytoplasm (*Kaplan and Tolwinski, 2010*). Thus, the polarized activity of Dsh must require distinct regulatory mechanisms at different sub-membrane compartments (*Mlodzik, 2016*). Our results provide the in vivo evidence suggesting that the basolateral polarity determinant Dlg1 may play a dominant role to control the Dsh abundance/activity in canonical Wg signaling.

Altered Dvl production or activity has been linked with several forms of cancer (*Kafka et al., 2014*). The stability of Dvl proteins can be controlled through regulated protein degradation both in vertebrates (*Gao and Chen, 2010*) and in *Drosophila* as reported in this study. In HEK293T cells, Dapper1 induces whilst Myc-interacting zinc-finger protein 1 (MIZ1) antagonizes autophagic degradation of Dvl2 in lysosome (*Zhang et al., 2006*; *Huang et al., 2015*; *Ma et al., 2015*). It is also reported that a tumor suppressor CYLD deubiquitinase inhibits the ubiquitination of Dvl (*Tauriello et al., 2010*). As Dlg1 prevents Dsh from degradation in *Drosophila*, it is important to investigate if Dlg1 participates in a posttranslational regulatory network of Dvl to integrate endocytosis and autophagy. Furthermore, upregulation of *dvl2* and *dlg2* expression has been found in various forms of cancer as shown in the COSMIC database (*Forbes et al., 2015*). The study of the interaction between Dlg1 and Dsh may aid the development of novel approaches to prevent or treat relevant diseases.

## Complex relationship between cell polarity determination and developmental signaling

Dlg1 acts together with L(2)gl to form a basolateral complex in polarized epithelium. Dsh is known to interact with L(2)gl. On one hand, Dsh activity is required for correct localization of L(2)gl to establish apical-basal polarity in *Xenopus* ectoderm and *Drosophila* follicular epithelium (*Dollar et al., 2005*). On the other hand, L(2)gl can regulate Dsh to maintain planar organization of the embryonic epidermis in *Drosophila* (*Kaplan and Tolwinski, 2010*). Despite the complex interaction between L(2)gl and Dsh, not much is known about mutual regulation between Dlg1 and Dsh. A recent report suggests that Dsh binds to Dlg1 to activate Guk Holder-dependent spindle positioning in *Drosophila* (*Garcia et al., 2014*). Our results unveil another side of the relationship in which Dlg1 controls the turnover of Dsh to ensure developmental signal propagation. Apart from its apical localization at the cell boundary, Dsh is also found in the basal side of the cytoplasm (*Kaplan and Tolwinski, 2010*). It is likely that the interactions among Dsh, Dlg1 and L(2)gl may be dependent on their localization, and Dsh may serve as a bridge to connect cell signaling and polarity.

Developmental signaling and cell polarity intertwine to control a diverse array of cellular events. It is well known that Wg/Wnt signaling controls cell polarity in distinct manner. Non-canonical signaling acts through cytoskeletal regulators to establish planar cell polarity (*Yang and Mlodzik, 2015*).

Canonical signaling may also directly affect apical-basal cell polarity (*Karner et al., 2006*). On the other hand, disruption of epithelial cell polarity has a profound impact on protein endocytosis and recycling (*Barbieri et al., 2016*), both of which are essential regulatory steps for signal activation and maintenance (*Shivas et al., 2010*). Our results add another layer of complexity by which polarity determinants could contribute to cell signaling independent of their conventional roles in polarity establishment and maintenance. Interestingly, this mode of regulation is also observed for other signaling processes. Loss of Dlg5 impairs Sonic hedgehog-induced Gli2 accumulation at the ciliary tip in mouse fibroblast cells that may not rely on cell polarity regulation (*Chong et al., 2015*). Similarly, L(2)gl regulates Notch signaling via endocytosis, independent of its role in cell polarity (*Parsons et al., 2014*). We believe that other cell polarity determinants may similarly participate in polarity-independent processes, however, the exact mechanism of how they cooperate to modulate developmental signaling awaits further investigation.

## Materials and methods

### Fly genetics

The following fly stocks were obtained from the Bloomington *Drosophila* Stock Center: *ap*-Gal4, *da*-Gal4, *dpp*-Gal4, *hh*-Gal4, *T80*-Gal4, *ts*-Gal80 (*McGuire et al., 2003*), *vg$^{BE}$*-Gal4, *baz$^4$ FRT9-2* (#23229), *cdc42$^4$ FRT19A* (#9106), *dlg1$^{14}$ FRT101* (#36283), *l(2)gl$^{27S3}$ FRT40A* (#41561), *Dll-lacZ* (#10981), *wg-lacZ* (#1672), UAS-*dsh* (#9524), UAS-*eIF4AIII RNAi* (#32444), UAS-*mago RNAi* (#28931), UAS-*Rnps1 RNAi* (#36580), and UAS-*tsu RNAi* (#28955). UAS-*eIF4AIII RNAi* (#108580), UAS-*mago RNAi* (#28132), UAS-*tsu RNAi* (#107385) and UAS-*Ubpy RNAi* (#107623) were obtained from the Vienna *Drosophila* RNAi Center (VDRC). UAS-*arr* was a gift of Xinhua Lin, *btz$^2$* (*Palacios and St Johnston, 2002*) was a gift of Daniel St Johnston, UAS-*dlg1-gfp* (*Zhang et al., 2007*) was a gift of Bingwei Lu, *dsh-gfp* (*Axelrod et al., 1998*) was a gift of Jeffrey Axelrod, *esg-flp* (*esg*-Gal4, UAS-*flp*; *Chen et al., 2005*) and *NRT-HA-wg* (*Alexandre et al., 2014*) were gifts of Jean-Paul Vincent, UAS-*fz2* (*Chen and Struhl, 1999*) was a gift of Gary Struhl, *fz3-lacZ* (*Sato et al., 1999*) was a gift of Roel Nusse, FRT42D *mago$^{93D}$* and FRT42D *tsu$^{\Delta18}$* (*Roignant and Treisman, 2010*) and FRT42D *M(2)58F ubi-gfp* were gifts of Jessica Treisman.

All fly crosses were maintained at 25°C unless noted otherwise. Detailed crossing schemes for each figure are shown in *Supplementary file 1*. Loss-of-function somatic clones were induced in the wing disc by Flp/FRT-mediated homologous recombination. Second instar larvae from parental crosses were heat-shocked at 37°C for half an hour. Phenotypes of the adult wing and wing discs are all fully penetrant (n>20).

### Immunofluorescence staining

For conventional immunofluorescence staining, wing discs dissected from third instar larvae were fixed in 4% paraformaldehyde, blocked in 0.2% BSA and incubated overnight at 4°C with the following primary antibodies: rabbit anti aPKC (1:500; sc-216; Santa Cruz Biotechnology, Dallas, TX), rabbit anti-Arr (1:15000; a gift of Stephen DiNardo; *Rives et al., 2006*), mouse anti β-galactosidase (1:50; 40-1a; Developmental Studies Hybridoma Bank, DSHB, Iowa City, IA), rabbit anti-β-galactosidase (1:4000; Cappel, Durham, NC), mouse anti-Dlg1 (1:50; 4F3; DSHB), mouse anti-Dll (1:400; a gift of Ian Duncan; *Duncan et al., 1998*), rat anti-DE-Cad (1:500; DCAD2; DSHB), rat anti-Dsh (1:500; a gift from Tadashi Uemura; *Shimada et al., 2001*), mouse anti-Fz2 (1:20; 12A7; DSHB), mouse anti-HA (1:100; 6E2, Cell Signaling Technology, CST, Danvers, MA), rabbit anti-LAMP1 (1:500; ab30687; abcam, Cambridge, MA; *Bouché et al., 2016*), rabbit anti-Rbsn-5 (1:2000; a gift of Akira Nakamura; *Tanaka and Nakamura, 2008*), guinea pig anti-Sens (1:1000; a gift of Hugo Bellen; *Nolo et al., 2000*) and mouse anti-Wg (1:200; 4D4; DSHB). The wing discs were incubated with Alexa fluor-conjugated secondary antibodies (1:400; Invitrogen, Carlsbad, CA) for one hour at room temperature before mounting. Fluorescence images were acquired with a Zeiss Axio Imager Z1 microscope equipped with an ApoTome or a Leica SP8 confocal microscope. The figures were assembled in Adobe Photoshop CS5. Minor image adjustments (brightness and/or contrast) were performed in AxioVision 4.8.1 or Photoshop.

Extracellular Wg staining was performed based on a previously described protocol with minor modification (*Strigini and Cohen, 2000*). Briefly, third instar larvae discs were dissected in ice-cold

Schneider's *Drosophila* medium (Invitrogen) supplemented with 10% FBS, 100 U/ml of penicillin and 100 mg/ml of streptomycin (full medium) and then incubated on ice with mouse anti-Wg antibody (1:10; 4D4) diluted in the full medium for one hour. Larval discs were rinsed and then fixed for 20 minutes in ice-cold PBS containing 4% paraformaldehyde before proceeding for immunofluorescence staining.

To detect Dsh protein that undergoes endocytic degradation in wing discs in which *dsh-gfp* expression was controlled by the *dsh* promoter, *Ubpy* RNAi was used to disrupt ESCRT-0 complex that is required for delivery of internalized cargos for lysosomal degradation (*Zhang et al., 2014*). Similarly, S2 cells overexpressing *dsh-Myc* was treated with chloroquine (10 mg/ml, Sigma, St. Louis, MO) to disrupt lysosome function. Antibodies against Rbsn-5 and LAMP1 were used to label early endosomes and late endosomes/lysosomes, respectively.

## Molecular biology

Myc-tagged fly *dsh* was generated by cloning the *dsh-Myc* fragment amplified from the UAS-*dsh* (Bloomington #9524) transgenic strain into the NotI/XbaI site of the pUAST vector (*Brand and Perrimon, 1993*) or a pCaSpeR-hs vector derived from pCaSpeR. *dsh-Flag* construct was generated by cutting full-length fly *dsh* cDNA from the pUAST vector and then subcloned into the EcoRI site of a pUAST-3×Flag vector. The HA-tagged fly *dlg1* was generated by fusing an HA tag at the N-terminus of the full-length *dlg1-RB* or *dlg1-RD* cDNA and then cloned into the pUAST vector. HA-tagged human *dvl1, dvl2* and *dvl3* were generated by fusing an HA tag at the C-terminus of the full-length human *dvl1, dvl2* and *dvl3* cDNA, respectively, and the resulting HA-fusions were then cloned into a pcDNA3.1 vector. Human *dlg* with a Flag tag at the C-terminus was generated by cloning the full-length *dlg2 or dlg3* cDNA into a pCMV3×Flag vector. This vector was also used to generate the *dlg2KKAA* or *dlg2SFI-NL* mutant expression vector by site directed mutagenesis (Stratagene, La Jolla, CA). pAct-*Myc-Ub* plasmid was provided by Shunsuke Ishii (*Dai et al., 2003*). Primers used in molecular cloning are listed in *Supplementary file 1*.

## Cell culture, transfection and RNAi treatment

*Drosophila* Schneider S2 cells were cultured at 25°C in Schneider's *Drosophila* full medium. HEK293T cells were grown in DMEM medium (Invitrogen) supplemented with 10% FBS, 100 µ/ml of penicillin and 100 mg/ml of streptomycin at 37°C. DNA transfection was carried out using a standard calcium phosphate protocol.

In some experiments, S2 cells or HEK293T cells were treated for up to four hours with cycloheximide (CHX; 50 µg/ml; Sigma) before harvest to inhibit nascent protein synthesis. MG132 (50 µM; Sigma) was used to inhibit the proteasome activity, while chloroquine (10 mg/ml) was used to inhibit lysosome function (*Zhang et al., 2014*). S2 cells transfected with *hs-dsh* and indicated vectors were heat shock for half an hour at 37°C after transfection for 48 hours. Then the cells were recovered at 25°C for one hour followed by drug treatment.

dsRNA was generated with the MEGAscript high yield transcription kit (Ambion, Austin, TX) according to the manufacturer's instruction. DNA template targeting *tsu* (encoding amino acids 45–165), *mago* (encoding amino acids 31–140), *dlg1* (encoding amino acids 800–917) and full length yeast *Gal80* was generated by PCR and used for dsRNA synthesis. dsRNA targeting yeast *Gal80* coding sequence was used as a negative control (*Su et al., 2011*). For RNAi knockdown in S2 cells, dsRNA transfection was carried out using a standard calcium phosphate protocol. Primers used to generate dsRNAs are listed in *Supplementary file 1*.

## TOPFlash Wnt signaling luciferase reporter assay

HEK293T cells grown in 24 well plates were transfected with the TOPFlash luciferase reporter (a gift of Yeguang Chen; *Zhang et al., 2006*) and indicated vectors for two days before harvest. The pRL-TK *Renilla* reporter was co-transfected to normalize transfection efficiency. Luciferase activity was measured following the Dual-Glo luciferase assay protocol (Promega, Madison, WI).

## Immunoblotting, immunoprecipitation and ubiquitination assays

S2, HEK293T cells and 3rdinstar larval wing imaginal discs were lysed in NP-40 buffer (1% NP-40, 150 mM NaCl and 50 mM Tris-Cl, pH 8) supplemented with protease inhibitor cocktail

(Roche, Germany). The concentration of protein cell lysate was quantified using a BCA protein assay kit (Thermo, Waltham, MA). Immunoblotting was carried out using standard protocols. The following antibodies were used for immunoblotting: mouse anti-β-Tubulin (1:10000; E7, DSHB), mouse anti-Cyclin B (1:50; F2F4; DSHB), mouse anti-Dlg1 (1:2000; 4F3; DSHB), rabbit anti-Flag tag (1:2000; D6W5B; CST), rabbit anti-HA tag (1:1000; Y-11; Santa Cruz), mouse anti-HA tag (1:2000; 6E2; CST), rabbit anti-MAPK (1:1000, 137F5, CST) and mouse anti-Myc tag (1:2000; 9B11; CST).

Immunoprecipitation was performed using agarose anti-HA (Vector Labs, Burlingame, CA), agarose anti-Myc (Vector Labs), agarose anti-GFP (Vector Labs) or Flag M2 affinity gel (Sigma) according to manufacturers' instructions. Immunoblots presented in all figures are representatives of at least three independent experiments.

Ubiquitination assays were carried out with hot lysis-extracted protein lysates based on the protocol described previously (*Row et al., 2006*; *Zhang et al., 2014*). Briefly, S2 cells transfected with *dsh-Flag, HA-dlg* and *6×Myc-Ub* were hot-lysed in denaturing buffer (1% SDS, 50 mM Tris, pH 7.5, 0.5 mM EDTA) by boiling for five minutes. Lysates were then diluted 10-fold with NP-40 lysis buffer and subject to immunoprecipitation with anti-Flag M2 affinity gel (Sigma).

## RNA isolation, quantitative real-time PCR and RT-PCR

Total RNA of pooled third instar larvae or imaginal wing discs was extracted using TRIzol reagent (Invitrogen). Residual genomic DNA was removed by RNase-free DNase (New England Biolabs, NEB, Ipswich, MA). First strand cDNA was synthesized using oligo-dT primers and SuperScript III reverse transcriptase (Invitrogen). Quantitative real-time PCR was performed using SYBR Green PCR master mix (Applied Biosystems, Waltham, MA) on a 7500 real time PCR system (Applied Biosystems). Using $\alpha$-*Tubulin 84B* as an internal control, relative fold changes of transcripts were calculated using comparative CT ($2^{-\Delta\Delta CT}$) method. Three independent samples were prepared and run in triplicates. RT-PCR was performed to compare with the quantitative real-time PCR results. Primers used in quantitative real-time PCR and RT-PCR are listed in *Supplementary file 1*.

## Whole transcriptome sequencing and analyses

Total RNA from pooled third instar larval wing imaginal discs (1000 pairs per sample preparation) expressing UAS-*lacZ* or UAS-*tsu RNAi* (VDRC#107385) driven by *T80*-Gal4 was extracted in duplicates using TRIzol. Poly(A)+ mRNAs were enriched using Dynabeads oligo (dT) beads (NEB). RT reactions and purification of cDNA templates were performed following the RNA-seq sample preparation protocol from Illumina. Each cDNA sample was sequenced on an IlluminaHiseq 2500 system.

Whole transcriptome reads were aligned using the TopHat (v2.0.13) (*Trapnell et al., 2012*) with the Ensembl Drosophila_melanogaster.BDGP5.78.gtf as a reference (http://www.ensembl.org/index. html). In total, 22,563,753 and 24,660,199 of 125 bp reads pairs for duplicated *tsu* samples and 22,610,210 and 28,651,178 of 125 bp reads pairs for *lacZ* samples were sequenced, respectively. The transcription analysis was performed using Cufflinks (v2.2.1) (*Trapnell et al., 2012*). DEXseq was used to plot the transcripts to each gene (*Anders et al., 2012*). RPKM method (reads per kilobase of transcript per million mapped sequence reads; *Mortazavi et al., 2008*) was used for normalizing gene counts. We calculated the ratio of RPKM between the *tsu* and *lacZ* samples. The R density plot package (*R Core Team, 2015*, R: A language and environment for statistical computing. R Foundation for Statistical Computing, Vienna, Austria. https://www.R-project.org//) was used to generate the distribution plot shown in *Figure 4—figure supplement 2C*. The Seq data was deposited to NCBI website: http://www.ensembl.org/index.html http://www.ncbi.nlm.nih.gov/geo/query/acc.cgi? token=ujyhqauybbujhun&acc=GSE81220.

## Acknowledgements

We thank everyone who shared with us reagents, and the Bloomington *Drosophila* Stock Center, the Vienna *Drosophila* RNAi center (VDRC) and the Developmental Studies Hybridoma Bank (DSHB) for fly stocks and antibodies.

## Additional information

### Funding

| Funder | Grant reference number | Author |
|---|---|---|
| State Key Laboratory of Membrane Biology of the People's Republic of China | | Alan Jian Zhu |
| Peking-Tsinghua Center for Life Sciences | | Cheng Li<br>Alan Jian Zhu |
| Ministry of Science and Technology of the People's Republic of China | 2014CB942804 | Alan Jian Zhu |
| National Science Foundation of the People's Republic of China | 31371410 | Alan Jian Zhu |
| National Science Foundation of the People's Republic of China | 31401241 | Min Liu |
| China Postdoctoral Science Foundation | Postdoctoral Fellowship, 2014M550556 | Min Liu |
| Peking-Tsinghua Center for Life Sciences | Postdoctoral Fellowship | Min Liu |

The funders had no role in study design, data collection and interpretation, or the decision to submit the work for publication.

### Author contributions

ML, Conception and design, Acquisition of data, Analysis and interpretation of data, Drafting or revising the article; YL, Acquisition of data, Analysis and interpretation of data, Drafting or revising the article; AL, YS, JD, Acquisition of data, Analysis and interpretation of data; RL, CL, Analysis and interpretation of data, Drafting or revising the article; AJZ, Conception and design, Analysis and interpretation of data, Drafting or revising the article

### Author ORCIDs

Alan Jian Zhu, http://orcid.org/0000-0001-8208-1729

## Additional files

### Supplementary files

• Supplementary file 1. Fly genetics information and Primers.

• Supplementary file 2. RNA-seq analyses identify candidate genes whose mRNA expression are subject to the pre-EJC regulation. Two criteria were used to determine candidate genes that are subject to the regulation by the pre-EJC: 1) the mRNA abundance values (measured by FPKM method) are above 10 in either *tsu* or *lacZ* samples (except for the value of *sens* that is listed at the bottom of the table), and 2) the expression levels are changed by more than 25% in the *tsu* sample. In total, 1447 genes were identified, in which the expression of 818 genes was downregulated.

• Supplementary file 3. Bioinformatic analyses deduce novel transcripts from the pre-EJC regulated candidate genes. 408 novel transcripts in 227 genes were identified by Cufflinks (v2.2.1).

### Major datasets

The following dataset was generated:

| Author(s) | Year | Dataset title | Dataset URL | Database, license, and accessibility information |
|---|---|---|---|---|
| Liu M, Li Y, Li R, Li C, Zhu AJ | 2016 | The exon junction complex regulates the splicing of cell polarity gene dlg1 to control Wingless signaling in development | http://www.ncbi.nlm.nih.gov/geo/query/acc.cgi?acc=GSE81220 | Publicly available at the NCBI Gene Expression Omnibus (accession no: GSE81220) |

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
