## [Decision Letter]

Thank you for submitting your article "The exon junction complex regulates the splicing of cell polarity gene dlg1 to control Wingless signaling in development" for consideration by *eLife*. Your article has been reviewed by three peer reviewers, one of whom is a member of our Board of Reviewing Editors, and the evaluation has been overseen by K VijayRaghavan as the Senior Editor. The reviewers have opted to remain anonymous.

The reviewers have discussed the reviews with one another and the Reviewing Editor has drafted this decision to help you prepare a revised submission.

Summary:

This paper reports a novel role for the exon junction complex (EJC) in regulating Wg signaling. Mechanistically, the authors shows that loss of EJC leads to inapproriate splicing and decreased protein level of dlg1. This in turn leads to reduced stability of Dsh, a Dlg1-binding protein and a key Wg signaling mediator. They also provide some evidence suggesting that this function of Dlg1 is independent from its well established role in cell polarity and that a mammalian orthologue of Dlg appears to play a conserved role in Wnt signaling by interacting with and stabilizing a mammalian homologue of Dsh.

Overall, this is a very interesting study that unveils a completely novel mechanism regulating Wg signaling. It is well written and well conducted for the most part. The reviewers have identified a few issues that need to be addressed in a revision.

Essential revisions:

1) In Figure 3, the reduction of endogenous Dsh protein levels in a tsu mutant background is simply not convincing. This is a central piece of data that supports the whole study. Other means documenting the impact of EJC loss-of-function on endogenous Dsh protein levels should be attempted to demonstrate this important point.

2) The authors may consider a few additional experiments to strengthen a causal link between EJC, Dlg and Dsh, as well as the specificity of the interactions.

2a) If dlg1 is the target of EJC in Wg signaling, the tsu or mago loss-of-function phenotypes should be rescued by Dlg1 overexpression. Furthermore, unlike endogenous Dlg1, the expression of this cDNA-derived Dlg1 protein should not be regulated by EJC.

2b) In Figure 3, it would be important to show that the mRNA levels of the Dsh-Myc construct (which uses a heterogenous promoter for its expression) is unaffected by EJC depletion.

2c) The links between the EJC, dlg1 and dsh could be strengthened by showing that Wg signaling is affected in dlg1 mutant clones and not only by dlg1 RNAi, which could have off-target effects.

3) Based on the analysis of l(2)gl, baz or cdc42 mutant clones, the authors conclude that the regulation of Dsh by Dlg1 is independent of Dlg1's function in cell polarity. It is important to show in this context that cell polarity is indeed disrupted in these mutants.

4) Do mago or Y14 clones show the altered cell polarity that would be predicted based on the reduction in Dlg1, or is it possible that the splice forms that are most dependent on the pre-EJC are dedicated to Dsh protection?

5) The immunoprecipitations in Figure 4 should include negative controls in which anti-Myc or anti-GFP is used to precipitate extracts that do not express the corresponding tagged form of Dsh, and similar controls should be added for immunoprecipitations in other figures.

6) In Figure 5, does chloroquine also stabilizes Dsh levels upon EJC depletion? This is important to support the notion that the same destabilization mechanism is at work on Dsh upon EJC or Dlg1 depletion. As lysosomal degradation of Dsh is unexpected, it would be preferable to strengthen this point by showing colocalization of Dsh with endosomal and lysosomal markers, rather than just by chloroquine treatment.

---

## [Author Response]

*Essential revisions:*

1) In Figure 3, the reduction of endogenous Dsh protein levels in a tsu mutant background is simply not convincing. This is a central piece of data that supports the whole study. Other means documenting the impact of EJC loss-of-function on endogenous Dsh protein levels should be attempted to demonstrate this important point.

Thanks for the reviewers to emphasize this important aspect of our experiments. We re-examined the expression patterns of endogenous Dsh protein in *tsu* as well as in *mago* loss-of-function mutant clones. In both cases, Dsh expression was obviously reduced (Figure 3).

*2) The authors may consider a few additional experiments to strengthen a causal link between EJC, Dlg and Dsh, as well as the specificity of the interactions.*

*2a) If dlg1 is the target of EJC in Wg signaling, the tsu or mago loss-of-function phenotypes should be rescued by Dlg1 overexpression. Furthermore, unlike endogenous Dlg1, the expression of this cDNA-derived Dlg1 protein should not be regulated by EJC.*

As anticipated by the reviewers, overexpressed *dlg1* largely rescued the wing margin defects caused by *tsu RNAi*. This new result is shown as Figure 4—figure supplement 3’. The pre-EJC acts at the level of *dlg1* mRNA splicing for Wg signaling. Consistent with this finding, HA-Dlg1 protein production derived from a cDNA expression construct (i.e. pUAST-*HA-dlg1*) was not altered when the expression of *tsu* or *mago* was reduced by RNAi in S2 cells. This result is now shown as Figure 4

*2b) In*
Figure 3*, it would be important to show that the mRNA levels of the Dsh-Myc construct (which uses a heterogenous promoter for its expression) is unaffected by EJC depletion.*

The EJC indirectly regulates the abundance of Dsh protein through its ability to maintain precise splicing of *dlg1*. We confirmed this finding by showing that the level of *dsh-Myc* mRNA was not altered when the expression of *tsu* or *mago* was knocked down by RNAi in S2 cells (Figure 3—figure supplement 2).

*2c) The links between the EJC, dlg1 and dsh could be strengthened by showing that Wg signaling is affected in dlg1 mutant clones and not only by dlg1 RNAi, which could have off-target effects.*

To avoid potential off-target effects of *dlg1 RNAi*, we re-examined Wg signaling activation using *dlg1^14^*allele. The activation of Sens was reduced in *dlg1* somatic mutant clones in wing discs (Figure 4—figure supplement 3). This result is consistent with what we observed in wing discs overexpressing *dlg1 RNAi* (Figure 4—figure supplement 3).

*3) Based on the analysis of l(2)gl, baz or cdc42 mutant clones, the authors conclude that the regulation of Dsh by Dlg1 is independent of Dlg1's function in cell polarity. It is important to show in this context that cell polarity is indeed disrupted in these mutants.*

To make sure that the polarity mutant alleles used in our study indeed affect cell polarity in wing discs, the localization pattern of DE-cadherin, a marker of the adherens junction, was examined. The localization of DE-Cadherin was obviously altered in polarity mutant clones in wing disc cells; its expression extended to more basolateral membrane domains (Figure 5—figure supplement 1).

*4) Do mago or Y14 clones show the altered cell polarity that would be predicted based on the reduction in Dlg1, or is it possible that the splice forms that are most dependent on the pre-EJC are dedicated to Dsh protection?*

We did observe altered expression of apical localized protein aPKC and adherens junction protein DE-Cadherin in *mago* and *tsu* mutant clones (Figure 4—figure supplement 5), suggesting that the EJC activity may affect both pools of Dlg1 that control Wg signaling (Dlg-Dsh interaction) as well as cell polarity (apical-basal antagonism).

5) The immunoprecipitations in Figure 4 should include negative controls in which anti-Myc or anti-GFP is used to precipitate extracts that do not express the corresponding tagged form of Dsh, and similar controls should be added for immunoprecipitations in other figures.

We used GFP alone and irrelevant proteins tagged with Myc or HA to perform the control co-IP experiments. These data are shown in Figure 4, and Figure 4—figure supplement 5.

6) In Figure 5, does chloroquine also stabilizes Dsh levels upon EJC depletion? This is important to support the notion that the same destabilization mechanism is at work on Dsh upon EJC or Dlg1 depletion. As lysosomal degradation of Dsh is unexpected, it would be preferable to strengthen this point by showing colocalization of Dsh with endosomal and lysosomal markers, rather than just by chloroquine treatment.

Disrupting lysosome function by treating S2 cells with chloroquine did stabilize Dsh protein when the expression of *tsu* was knocked down by RNAi (Figure 5). Consistently, both endogenous Dsh in wing discs (Figure 5) and Dsh-Myc in S2 cells (Figure 5—figure supplement 4) were found to colocalize with early endosome marker Rbsn-5 and late endosome/lysosome marker LAMP1 when the lysosomal degradation pathway was blocked.